# Chitosan Coatings Modified with Nanostructured ZnO for the Preservation of Strawberries

**DOI:** 10.3390/polym15183772

**Published:** 2023-09-15

**Authors:** Dulce J. García-García, G. F. Pérez-Sánchez, H. Hernández-Cocoletzi, M. G. Sánchez-Arzubide, M. L. Luna-Guevara, E. Rubio-Rosas, Rambabu Krishnamoorthy, C. Morán-Raya

**Affiliations:** 1Ecocampus Valsequillo, ICUAP, Centro de Investigación en Fisicoquímica de Materiales, Benemérita Universidad Autónoma de Puebla, Edificio Val-3, San Pedro Zacachimapa, Puebla 72960, Mexico; dulce.garciagarcia@alumno.buap.mx (D.J.G.-G.); carolina.moran@correo.buap.mx (C.M.-R.); 2Facultad de Ingeniería Química, Benemérita Universidad Autónoma de Puebla, Av. San Claudio y 18 sur S/N Edificio FIQ7 CU San Manuel, Puebla 72570, Mexico; madai.sancheza@correo.buap.mx (M.G.S.-A.); maria.luna@correo.buap.mx (M.L.L.-G.); 3Centro Universitario de Vinculación y Transferencia de Tecnología, Benemérita Universidad Autónoma de Puebla, Prol. 24 sur S/N CU San Manuel, Puebla 72570, Mexico; efrain.rubio@correo.buap.mx; 4Department of Chemical Engineering, Khalifa University, Abu Dhabi P.O. Box 127788, United Arab Emirates

**Keywords:** strawberry, food preservation, chitosan, ZnO nanoparticles, composites, edible coatings

## Abstract

Strawberries are highly consumed around the world; however, the post-harvest shelf life is a market challenge to mitigate. It is necessary to guarantee the taste, color, and nutritional value of the fruit for a prolonged period of time. In this work, a nanocoating based on chitosan and ZnO nanoparticles for the preservation of strawberries was developed and examined. The chitosan was obtained from residual shrimp skeletons using the chemical method, and the ZnO nanoparticles were synthesized by the close-spaced sublimation method. X-ray diffraction, scanning electron microscopy, electron dispersion analysis, transmission electron microscopy, and infrared spectroscopy were used to characterize the hybrid coating. The spaghetti-like ZnO nanoparticles presented the typical wurtzite structure, which was uniformly distributed into the chitosan matrix, as observed by the elemental mapping. Measurements of color, texture, pH, titratable acidity, humidity content, and microbiological tests were performed for the strawberries coated with the Chitosan/ZnO hybrid coating, which was uniformly impregnated on the strawberries’ surface. After eight days of storage, the fruit maintained a fresh appearance. The microbial load was reduced because of the synergistic effect between chitosan and ZnO nanoparticles. Global results confirm that coated strawberries are suitable for human consumption.

## 1. Introduction

Strawberries are very attractive for their flavor and antioxidant content. They are perishable, with a preservation time dependent on the storage procedure. The damage to the strawberries is commonly due to incorrect handling and storage; during these steps, biological, chemical, and physical factors influence their quality. The production process involves the harvest, the transport, the load, and the download. In these steps, strawberries are exposed to dangerous biological agents, reducing their shelf life [1] and causing an economic loss of about 30% to growers and marketers.

Zinc oxide nanoparticles (ZnO-NPs) have attracted attention due to their antimicrobial properties, low cost of production, and low toxicity [2]. It is possible to obtain ZnO-NPs with a variety of chemical and physical methods in different shapes and sizes. In recent years, environmentally friendly methods (green synthesis) have been developed using different plant extracts for the preparation and characterization of chitosan–nano-ZnO composite films for the preservation of cherry tomatoes [3]. It has been demonstrated that nanostructured ZnO can inhibit the growth of bacteria and fungi, extending the shelf life of foods [4]. Unlike other metal oxides, it is recognized as a safe material by the Food and Drug Administration (FDA) in the US [5]. However, to date, there are no conclusive results; in an in vitro colon simulation, it was reported that nano-ZnO could alter the metabolism, microbiota, and resistome of the human gut [6]. Even though nanostructured ZnO has been used in food, medicine, and feed production and processing, it is generally inefficient to improve the antioxidant properties of edible films [7].

Chitosan is a linear polysaccharide scarcely found in nature. This biopolymer is obtained from chitin through a deacetylation reaction [8]. The main source of chitin is shrimp, lobster, and crab skeletons. Chitosan has been widely used in different areas, such as agriculture, medicine, food, cosmetics, textiles, pharmaceuticals, biotechnology, and wastewater treatment [9]. In the food industry, it is widely used as an edible semipermeable barrier; however, its mechanical properties reduce its performance in protecting fruits. The effectiveness of chitosan can be enhanced when combined with organic and inorganic compounds. Being loaded with metal oxides such as zinc oxide and titanium oxide, it is possible to improve firmness, maintain quality, extend post-harvest life, and reduce pesticide residues in fruits [10].

Nanobiocomposites open up an opportunity for the use of innovative, high-performance, lightweight, and ecological composite materials, which makes them ideal materials to replace traditional non-biodegradable plastics. Nanostructured ZnO is highly viable for developing composite materials because it has high antimicrobial activity and improved mechanical and barrier properties [11]. The antibacterial mechanism associated with ZnO nanoparticles depends both on the generation of reactive oxygen species and on the release of antimicrobial Zinc ions; likewise, it has been shown that its effectiveness increases as the size of the particles decreases due to the increase of its surface reactivity [11,12]. Chitosan/ZnO nanocomposites have attracted the attention of researchers due to their antibacterial activity [13]. According to the surveys, Chitosan/ZnO bionanocomposites are limited to films [3,14]. Al-naamani et al. [15] reported the method for preparing PE films coated with chitosan ZnO composites with effective antimicrobial effects, and Karkar et al. [16] developed a film with Nigella sativa extract for preventing grape spoilage. Chitosan/ZnO coatings have been recently applied to extend the shelf life of wild-simulated Korean ginseng root [17] as well as inhibit microbial growth on fresh-cut papaya [18]. More complex Chitosan/ZnO nanocoatings have been proposed, including compounds such as alginate, which were applied to the preservation of guavas [19].

Concerning strawberries, the studies developed to date are limited to investigating the antibacterial activity of the coatings [20]. However, studies are required in which the quality of the fruit during storage is evaluated and the physicochemical parameters related to the level of consumer acceptance are maintained. In addition, preserving fruits that contain antioxidant properties associated with human health is required. Environmentally friendly and easily applicable approaches for preserving strawberries during storage are required. Chitosan ZnO coatings are promising materials to be used in the food industry; however, more studies are required. Thus, this work aims to develop a bionanocoating containing inorganic nanostructured ZnO and the natural biopolymer chitosan with the potential to be used in the preservation of fruits. This is demonstrated by different microbiological (Aerobic Mesophilic Bacteria, molds, and yeasts) as well as physicochemical (moisture content, texture, pH, soluble solids, titratable acidity, and color) analyses.

## 2. Materials and Methods

### 2.1. Chitosan Synthesis

Chitosan (Ch) was prepared using a methodology described in our previous work [21]. Briefly, raw shrimp shells were obtained from a local seafood restaurant; heads and ends were not included. After washing with distilled water and drying at 90 °C for 3 h, the shells were pulverized to a particle size in the range of 44–53 µm. The as-obtained sample was demineralized using 0.6 M HCl with a ratio of dried shells to an acid solution of 1:11 (*w*/*v*) for 3 h at 30 °C and stirring at 300 rpm. Afterwards, to obtain chitin, the sample was sonicated with a high-frequency ultrasonic bath in deionized water for 40 min. Finally, NaOH (50%) was added to chitin in the proportion 1:4 (*w*/*v*) with a constant stirring speed of 700 rpm. It was initially heated and maintained at 70 °C for 2 h, followed by heating at 115 °C for another 2 h. The final product of each step was washed with distilled water until it became neutral and then dried at 90 °C for 3 h. The as-obtained chitosan has a molecular weight of 56 kDa and a deacetylation degree of 94%.

### 2.2. Infrared Spectroscopy Characterization

The identification of chitosan was made using infrared spectroscopy. A Perkin Elmer Spectrophotometer with a fast Fourier transform and ATR system was employed for this purpose. The scan was carried out in the range of 4000–500 cm^−1^, employing eight scans with a 2 cm^−1^ resolution. Humidity (0%) and a temperature of 25 °C were maintained while obtaining the spectra to ensure proper determination of the –OH groups. Prior to the analysis, the samples were dried in a conventional heater at 105 °C for 4 h, and about 0.05 g of the sample was subjected to the analysis.

### 2.3. Synthesis of Nanostructured ZnO Using the Close-Spaced Sublimation Method

The zinc oxide nanostructures were synthesized using the close-space sublimation (CSS) method on glass substrates in two simple steps: sublimation of metallic zinc followed by thermal annealing at atmospheric pressure conditions. The experimental procedure is briefly described as follows: glass substrates of 2 cm^2^ were cleaned sequentially with xylene, acetone, and propanol at 50 °C for 10 min in an ultrasonic cleaner; finally, they were rinsed with deionized water (18 MΩ-cm) and dried with high-purity nitrogen. For the synthesis of the zinc nanostructures by CSS, initially, circular pellets made from zinc powders (99%, mesh size −325 microns) were used as the source. Inside the CSS system, the zinc pellet was placed on a graphite heater, establishing a distance separation between the source and the substrate of 1 cm by means of a quartz ring (thickness 0.76 mm). The growth deposition parameters of the zinc nanostructured films were fixed at 350 °C, 10 min, and 5 × 10^−3^ torr for the reactor temperature, growth time, and vacuum, respectively. The as-grown films showed an opaque gray color (to the naked eye). Finally, the zinc nanostructures were oxidized in the air using a tubular furnace at a temperature of 500 °C for one hour at atmospheric pressure. At the end of the treatment, the films turned white, indicating a conversion of zinc-to-zinc oxide nanostructures. The procedure for the HR-TEM analysis consisted of the mechanical separation of ZnO nanostructured material from the substrates after thermal annealing in air. After that, 4 mg of the as-prepared ZnO powder were dispersed in 50 mL of ethanol solvent, and the solution was homogenized in a bath sonicator for 30 min. Then, a drop of the solution was cast on a copper grid (400 mesh size) and dried on a hotplate at 70 °C for less than a minute. Finally, the grid was introduced into the HR-TEM system, and measurements were performed using a voltage acceleration of 200 KV.

### 2.4. Preparation and Application of the Coating

A solution with 1 mL of acetic acid 1 M and 100 mL of water was prepared; after homogenization, 1 g of chitosan was dissolved into this solution at 80 °C for 30 min. A second solution was prepared under the same conditions; however, in this case, 0.15 g ZnO nanoparticles were incorporated with constant stirring for 10 min. The obtained solutions were applied by submerging the strawberries for 1 and 2 s. It is important to mention that the coating was firmly adhered to the strawberry surface with insignificant runoff. A strawberry without any coating was used as a control sample. All the experiments were performed at room temperature and in triplicate. The samples were stored at 5 °C and observed for eight days [22].

### 2.5. Characterization of the Coating

The solution used for coating strawberries was analyzed by FTIR spectroscopy. The coating, once applied to the strawberries, was characterized by scanning electron microscopy (SEM) (JEOL JSM-6610LV, Tokyo, Japan); an elemental and a mapping analysis were developed to identify the main components of the coating. The accelerating voltage was 20 kV in both morphological characterizations and EDX measurements. A 2 × 2 cm portion of the strawberry surface was cut and heated at 40 °C for 4 h; the as obtained sample was Au covered. The mapping analysis was developed in a 3 × 3 mm area in 3 different zones; all the image fields were considered.

### 2.6. Chitosan and Ch/ZnO-NPS Coating Functionality in the Preservation of Strawberries

#### 2.6.1. Colorimetric Test

Strawberries with uniform size and commercial maturity were selected for the preservation study. They were purchased at a local market and used without any additional treatment. The sample luminosity was measured with a Hunter lab colorimeter every 24 h for eight days. With this parameter, the chroma (*C_ab_*^*^) for strawberries was obtained using Equation (1).
(1)$$the\;C_{ab}^{*}=\sqrt{(a^{*2}+b^{*2}}$$
where *a*^*^ represents the red and green colors; *b*^*^ the yellow and blue colors. The browning index (*IP*) was calculated with Equation (2) [23].
(2)$$IP=\frac{100\times \left(X-0.31\right)}{0.172}$$
where
(3)$$X=\frac{a^{*}+1.75L}{5.645L+a^{*}-3.012b^{*}}$$

#### 2.6.2. Texture Test

The texture test was performed with a TA.XT2i (Stable Micro Systems) texture analyzer. For that, a plane cylindrical probe p/3 (3 mm diameter) was used with a speed of 1 mm/s and a resistance time of 5 s for the analysis.

#### 2.6.3. pH, Titratable Acidity, Soluble Solids, and Humidity Content Determination

The pH, the soluble solids, and the titratable acidity were measured according to the AOAC 981.12, AOAC 932.12, and AOAC 942.15 methods [24], respectively. The pH of ground strawberries was measured with a WPA CD310 potentiometer. The titratable acidity was measured with titration equipment. The titration solution (8.9 mL) was prepared with NaOH at 0.1 N. For developing the titration, 10 mL of ground strawberries were incorporated into 10 mL of purified water; after mixing, two drops of phenolphthalein were added and mixed again. The obtained sample was titrated by adding two or three drops of the titration solution until observing a light pink color. The volumes of the NaOH solution were used for calculating the titratable acidity. For the determination of soluble solids content, strawberries were ground until they obtained a homogeneous mix. Each sample was placed in a digital refractometer (Atago, RX-100, Bellevue, DC, USA), and the measurements were expressed in °Bx. The humidity content was determined with the gravimetric method established in 934.06 of the AOAC standard. Each sample previously weighed was dried at 105 °C for 42 h in a drying oven; after that, the weight was determined. The humidity percentage was obtained with Equation (4).
(4)$$\%\;humidity=\frac{M_{0}-M_{f}}{M_{0}}\times 100$$
where *M*_0_ is the wet sample mass (g), and *M_f_* is the dry sample mass (g).

#### 2.6.4. Aerobic Mesophilic Bacteria, Fungi, and Yeast

For the microbiological tests, 10 g of each sample was introduced into sterile bags and homogenized for 1 min. The aerobic mesophilic bacteria (AMB) were measured with the plate pouring method on standard count agar; the plates were incubated at 35 °C for 48 h. The fungi and yeast quantification was realized by employing the plate pouring method on potato dextrose agar (PDA) acidified with 10% tartaric acid at a pH of 3.5; the plates were incubated at 25 °C for 72–120 h. The results were expressed as colony-forming units (CFU/g).

### 2.7. Statistical Analysis

All experiments and measurements were performed in triplicate, using a fully random design. An analysis of variance (ANOVA) of simple classification was applied, and the Tukey test was used to determine the difference between the means. The data analysis was carried out with Statistics Plus software (Statgraphics Centurion 19, Statistical Graphics Corp., Manugistics, Inc., Cambridge, MA, USA). A *p* ≤ 0.05 significance level was established for all cases.

## 3. Results

### 3.1. X-ray Diffraction

Figure 1 shows the XRD pattern of the spaghetti-like ZnO nanoparticles (ZnO-NPs) synthesized by thermal annealing-assisted CSS. The diffraction peaks located at 2θ = 31.7, 34.4, 36.2, 47.5, 56.5, 62.8, 66.3, 67.9, and 68.9° were indexed to the planes (100), (002), (101), (102), (110), (103), (200), (112), and (201), which are characteristic of the wurtzite phase of ZnO in accordance with JCPDS file number 36-1451; no signs of secondary phases or impurities were observed, thus confirming the crystalline nature of the ZnO NPs powder [25]. The crystalline size of ZnO nanostructures was calculated using the Williamson-Hall (W-H) method using the software provided by the XRD system (High Score Plus for Crystallite Size Analysis). The W-H analysis is an integral breadth method where both size and strain-induced broadening are considered in the deconvolution of the peak versus 2θ [26]. It is important to mention that in this research only the crystallite size was considered because the W-H plot is more realistic than using the Debye-Scherrer equation; moreover, the result estimated from this analysis was in good agreement with the corresponding HR-TEM results; the size obtained was 40.1 nm. Similar values of crystal size (46 nm) were found in pure ZnO nanoparticles [27]; however, they were slightly higher than those obtained in ZnO nanoparticles with chitosan [24,27] and ZnO added with citrus extracts [28].

### 3.2. Scanning Electron Microscopy

The SEM images of the as-synthesized nanostructured Zn by the CSS method are shown in Figure 2. The image at low magnification (Figure 2a) shows a homogeneous distribution of particle agglomerates with spaghetti-like morphology; their lengths reach approximately 5 μm, with a cross-section between 100 and 300 nm (Figure 2b,c). Figure 3 shows SEM images of the nanostructured ZnO layers (ZnO-NPs) obtained after the thermal treatment of Zn. In Figure 3a, the initial morphology of zinc nanostructures was not modified by the thermal stress. However, it could be observed after this process that the surface of the spaghetti-like nanostructures became rougher, along with the appearance of nanostructures of smaller dimensions on the surface (Figure 3b,c) and comparable results were reported elsewhere [29].

### 3.3. Transmission Electron Microscopy

TEM images of three different regions on the surface of ZnO nanostructures are shown in Figure 4. Figure 4a shows that agglomerates whose sizes are between 100 and 300 nm form ZnO nanostructures. On the other hand, Figure 4b reveals to us that the agglomerates are made up of nanoparticles whose average size corresponds to 50 nm, which is consistent with the previous XRD results. In addition, these nanoparticles tend to coalesce because of thermal annealing, forming nanorods that together give rise to the spaghetti-like morphology (Figure 4c), as observed in the SEM analysis. The agglomeration of nanoparticles is due to the densification process because of the narrow space between the metallic anions [30], giving rise to the formation of nanowires whose cross section is around 200 nm.

In general, the nanostructured ZnO prepared in this work showed higher values in nanoparticle size than those obtained in nanoparticles added with natural extracts (14–24 nm) [30,31] but like those obtained in commercial nanomaterials (50–60 nm) [14]. Finally, it was observed that there is a good correspondence between the results obtained by the X-ray diffraction and TEM techniques, given that in both cases, a crystallite size of ≈40 nm was obtained.

### 3.4. Fourier Transform Infrared Spectroscopy Analysis

Figure 5 shows the FTIR spectra of chitosan and Ch/ZnO-NPs bionanocomposite. The spectrum contains the characteristic absorption bands of chitosan at 3356 and 3398 cm^−1^, which correspond to the stretching vibrations of the OH groups. At 2877 cm^−1^, the vibrations attributed to the CH_2_ groups are shown. At 1647 cm^−1^, the bands corresponding to the bending vibrations from the N-H bonds and the stretching of the C-O bonds of the amine and amide groups are observed. The signals at ~1585 cm^−1^ are attributed to the vibration of the C=O bonds of the amide group [32]. At 1419 cm^−1^ the signals related to the CH_2_ bonds are observed, and ~1377 cm^−1^, the bands attributed to the C-O bond of the primary alcoholic group in the chitosan structure are observed. From 1060 to 1250 cm^−1^, the bending vibrations of the C-O-C groups of glucose were identified; around 1026 cm^−1^, the presence of the free amine groups (-NH_2_) of glucosamine is observed. Finally, at 575 cm^−1^ the bending vibrations of the NH bonds are identified. According to these results, it is concluded that the chitosan obtained has the characteristic bands reported in the literature [33].

The FTIR spectrum of the Ch/ZnO-NPs bio-nanocomposite is also shown in Figure 5. A spectrum such as that of chitosan is observed; the vibrations located between 500 and 800 cm^−1^ correspond to the stretching of the O-Zn-O bonds, which are not present in the chitosan spectrum, confirming the presence of ZnO in the nanocomposite [27,34].

### 3.5. Energy Dispersive Spectroscopy Analysis

Elemental mapping of strawberries coated with chitosan during 1 min of immersion (FRCh-1) is shown in Figure 6. The spectrum revealed that the chitosan coating was composed of C (33.8%) and O (52.5%), whose distribution is seen uniformly on the strawberry surface [35]. The chemical composition of the Ch/ZnO-NPs coated strawberry immersed for 1 min immersion (FRBN-1) is presented in Figure 7. In addition to C (35.70%) and O (54.77%), a homogeneous distribution of Zn (0.56%) (Figure 7i) was found on the strawberry surface without severe aggregation [26,36]. Moreover, there is no evidence of the lump’s formation, as has been observed in other cases where ZnO nanoparticles were used [16]. The homogeneous distribution of the Zn, as shown in Figure 7i, permits the inference that the ZnO nanostructures are uniformly distributed on the coating, which indicates a homogenous dispersion and integration into the chitosan matrix [26,36]. This also suggests high compatibility between the ZnO-NPs and the chitosan matrix due to the interactions between the free hydroxyl groups of chitosan and the available ZnO-NPs in the composite [37]. Thus, improved properties for the Ch/ZnO-NPs coating are expected for a longer shelf life of strawberries [38].

### 3.6. Coating Thickness

The thickness of the coating was measured from SEM images (Figure 8). For this, cross-sections of both pristine and ZnO nanocomposite coatings were prepared. The sample constituted by the chitosan coating immersed for 1 min (FRCh-1) presents an apparent longitudinal uniformity with an average thickness of 2.2 μm (Figure 8a). On the contrary, the thickness obtained after 2 min of immersion (FRCh-2) does not present uniformity in thickness, and their values are in the range of 2–3 μm (Figure 8b). In the case of Ch/ZnO NPs coating with an immersion time of 1 min (FRBN-1), SEM images at higher magnifications evidenced the appearance of randomly distributed particles on the surface of the coated strawberry, which is presumably an indication that the nanoparticles are incorporated throughout the chitosan matrix volume; the thickness was found in the range of 2–3 μm (Figure 8c). For the sample with 2 min of immersion (FRBN-2), it was observed that the cross section was rougher compared to the pristine coating, reinforcing the hypothesis that the ZnO nanoparticles lie on the surface as well as in the volume of the coating. The thickness does not differ significantly from the previous case, 2–3 μm (Figure 8d); therefore, we can conclude that the immersion times used in these experiments do not represent a notable change in the thickness of the coatings.

### 3.7. Functionality of the Ch/ZnO-NSs Coating in the Preservation of Strawberries

Strawberries with and without coating were stored at 5 °C and 25 °C for 8 days (Table 1 and Table 2). Table 1 shows that the microbial spoilage on the surface was lower in the strawberries coated with the Ch/ZnO-NPs nanobiocomposite than in the control strawberries; this is attributed to the antimicrobial properties of both chitosan and nanostructured ZnO. Cold storage reduces respiration rate and moisture loss, retarding microbial growth, allowing the shelf life to be extended and the quality of fruits to be preserved [39].

Table 2 shows that during storage of the coated strawberries (FRCh-1 and FRBN-1) at room temperature, they did not present microbial growth on the surface compared to the control strawberries. However, the coated strawberries have rough tissue on the surface, which suggests moisture loss during storage, which can give the appearance of deterioration. Table 1 and Table 2 reveal that the coatings prepared in this work retarded weight loss and firmness for 8 days, attributed to their barrier properties. Films prepared with PVA and TP preserved strawberries for 5 days [40], while LDPE filled with LAE films extended the shelf life to 10 days [41].

The antifungal activity of FRBN-1 could be attributed to the previously identified action of zinc oxide. It has been reported that the antifungal activity of ZnO can be attributed to the generation of intracellular reactive oxygen species (ROS) by the nanobiocomposite in direct contact with the fungal cell wall [42,43]. The generated ROS results in elevated stress, leading to oxidative damage to the fungal cell wall and cellular components. Previous reports suggested that the antifungal activity of ZnO NPs was mediated by ROS production [44]. There exists a synergistic or complementary effect between the ZnO and the chitosan; the resulting synergetic interactions were effective against other pathogen microorganisms [45].

Visually, the presence of mold, the generation of fermentative odors, and the loss of turgidity and firmness were observed in strawberries subjected to both temperatures on day 1 and day 8 for the FRCh-1 treatment but not for the FRBN-1 treatment. The presence of fungi is more pronounced in the neat Ch coating (FRCh-1) compared to the Ch/ZnO coating (FRBN-1). This suggests that the incorporation of nanoZnO into the chitosan coating has a positive effect on inhibiting the presence of fungi in the strawberries. This observation indicates that the Ch/ZnO coating is more effective in preventing fungal growth compared to the neat chitosan coating.

### 3.8. Microbiological Tests

The results of the microbiological analysis (Aerobic Mesophilic Bacteria, molds, and yeasts) of the strawberries stored for 8 days at 5 and 25 °C are shown in Table 3. A progressive and significant increase in colonies of aerobic mesophilic bacteria (AMB) was observed in the control samples with storage time, mainly in the refrigerated strawberries. Coated strawberries under refrigeration (5 °C) also presented an increase in the AMB; this is expected because most of the AMB has psychotropic ability and may grow in low-temperature environments. The initial microbial load of the strawberry would also have contributed to the result obtained. As can be seen, the strawberries coated with chitosan follow the same trend. A better performance was obtained with the Ch/ZnO-NPs coating; despite the fact that the strawberries under refrigeration presented an increase in the AMB (in the order of 10^5^), they are suitable for human consumption [31]. It is observed an increase in mold colonies and yeast in the control samples, mainly in those refrigerated. The FRCh-coated strawberries presented molds and yeast but in a reduced quantity; this confirmed the inhibition efficacy of chitosan against microorganisms on strawberries; the same effect has also been observed in tomatoes and ginseng [30,45].

The incorporation of the nanostructured ZnO enhanced the inhibition capability of the composite; at room temperature (25 °C), no fungus or yeast was found after 8 days of storage. These results confirm the effectiveness of ZnO application as an antifungal agent incorporated into chitosan for preserving strawberries. This effect has also been observed in fresh foods, such as orange juice [46,47]. Likewise, it has been shown that the incorporation of nanomaterials in edible polymer coatings improves their physical properties, such as oxygen and moisture barrier properties, which inhibit the growth of microorganisms in coated fresh fruits [48].

### 3.9. Moisture Content

Table 4 shows the moisture content of the samples stored for 8 days (5 and 25 °C). The control sample presented a higher reduction in humidity. Coatings have the effect of maintaining the moisture of strawberries; better performance was reached under refrigeration, where this parameter decreased no significantly. At room temperature, a reduction of less than 10% in this parameter was shown in the coated strawberries. The coating solutions created a barrier on the surface of the strawberries that prevented moisture loss. This effect has also been reported in sodium alginate coatings, where the protective layer between fresh fruits and the surrounding atmosphere decreases moisture transfer and the exchange of O_2_ and CO_2_ [49].

It should be noted that the moisture content was reduced when refrigeration conditions (5 °C) were used for storage. The application of low temperatures with complementary methods of food preservation, such as chemical treatments and edible coatings, generates lower moisture losses in the applied foods. It has been shown that the use of ZnO nanoparticles together with alginate significantly reduced (*p* < 0.05) the weight loss of burs [50].

### 3.10. Texture Test Analysis

Texture is one of the most important attributes for consumers when evaluating the quality of fruits. Firmness changes of strawberries with different coatings are shown in Figure 9; they were determined on Day 1 and Day 8 of storage. Firmness values obtained at the beginning of storage were 2.2 N (25 °C) and 0.6 N (5 °C). At refrigeration, the firmness value of most of the coated strawberries was lower than the control strawberry, except for those coated with the nanocomposite with an immersion time of 2 s (FRBN-2). The samples subjected to storage at room temperature (25 °C) presented a loss of firmness, impeding the measurement. Finally, the firmness of control strawberries showed significant differences (*p* < 0.05) compared to coated strawberry samples during cold storage.

In general, the control and coated strawberries showed a significant decrease in firmness. This decrease is due to the greater migration of water vapor on the surface of the fruit, which favors the growth of different fungi (*Botrytis cinera* and *Rhizopus stolonifer*). Both molds cause structural damage to the tissues and allow their softening [38]. Moreover, the decrease in firmness is related to the increase in moisture loss. In addition, the decrease in firmness which was observed during the first eight days of storage in the different evaluated coatings could be related to the degradation of the cortical parenchyma that forms the cell wall due to enzymatic degradation processes and to the same moisture loss during storage [39,44].

### 3.11. pH, Soluble Solids, and Titratable Acidity Analysis

Sweetness and acidity are among the essential indices for evaluating the flavor of fruits, which are evaluated by physicochemical parameters such as pH, total soluble solids (TSS), and titratable acidity (TA) [21]. Table 5 shows these values evaluated for coated and uncoated strawberries stored under refrigeration (5 °C) and at room temperature (25 °C). It is important to mention that these parameters can be influenced by factors such as crops, agricultural practices, agricultural region, season, etc. [51]. The pH of the uncoated samples increased slightly during storage, while no significant differences were observed in the coated samples (Table 5); similar results were reported by Nunes et al. (2006) [52].

Regarding TSS and TA, a decreasing trend was observed in the strawberries stored at room temperature. This means that the coatings control the strawberries’ maturity, preventing an increase in the TSS [46,52]. The variations of TA and TSS values in coated and uncoated strawberries during storage were not statistically significant (*p* > 0.05). A tendency to decrease in acidity and TSS was observed with the increase in pH of strawberries, possibly because, in the case of edible coatings, they slow down the respiratory rate of strawberries and delay the utilization of organic acids in enzymatic reactions [53]. The obtained results coincide with those of different studies on the application of chitosan-based coatings [38,54]. The values obtained for the coated strawberries are comparable with the parameters of both quality and commercial acceptability of fresh strawberries (pH = 3.17–3.98; TA = 0.60–0.82%; TSS = 6.7–10.3°Bx) [55].

### 3.12. Color

Color is a critical parameter that affects the customer’s selection and purchase decision for most fruits. Table 6 contains the color parameters *L*, *a**, *b**, and browning index (*PI*) of strawberries with and without coating during storage. As can be seen in Table 6, there is a significant difference in L in the different coatings, which is attributed to the process of oxidation and moisture loss that the strawberries suffered during storage [56,57,58].

Finally, it was found that the application of the coatings did not affect the luminosity of the strawberries with respect to the control sample. Regarding the chromatic coordinate *a** (the reddish hue of the strawberry epidermis), it did not show a statistical difference either at the beginning or at the end of storage; the measurements behaved as a homogeneous group. Looking at the data from Day-8 samples, a notable increase and subsequent decrease in the *a** values are observed, with a statistically significant difference. The increase in the shade of *a** is due to moisture loss during storage due to transpiration; the decrease in redness is probably due to an increase in respiratory and enzymatic activity that causes loss of quality due to oxidative browning [58].

Likewise, between the beginning and Day 8 of storage, no statistical difference was found (*p* > 0.05) between coated strawberries with respect to the chromatic coordinate *b** (the yellow hue of the strawberry’s epidermis). On Day 8, a decrease in *b** was observed in the coated strawberries compared to the control; this is associated with enzymatic browning reactions [39,54,55]. Finally, the browning index values obtained were found to be in the range of 97–117 and 50–152 for Day-1 and Day-8 samples, respectively, which indicates an increasing trend with storage time. The results indicate that, despite the chitosan and zinc oxide coatings acting as a selective barrier that prevents the fruit from being exposed to ambient oxygen, they present possible oxidation reactions [53], as well as a decrease in ascorbic acid due to its degradation over time, which promotes enzymatic-browning reactions [57,58].

It should be noted that on Day 8, the browning index values for the Ch/ZnO-NPs-1 coating at temperatures of 5 and 25 °C were comparable to those obtained with the control treatment. In general, both the coated and the control strawberries presented statistical differences on day 8 with respect to the beginning (day 0), due to the effect of time, differences that are attributed to moisture loss throughout the storage period. This moisture loss is assumed to be caused by biological phenomena typical of plant tissues, such as transpiration [58]. In this regard, various authors mention that chitosan-based coatings can delay external color changes in strawberries, similar to other edible coatings based on natural biopolymers [59].

## 4. Conclusions

In this study, a nanocoating based on chitosan and nanostructured spaghetti-like ZnO nanostructures (Ch/ZnO-NPs) was developed. The coating showed important microbiological and physicochemical property enhancements for strawberry preservation. The wurtzite ZnO-NPs are spaghetti-like, with a crystal size of 40 nm. The Ch/ZnO-NPs coating, with a thickness of 2–3 μm, was uniformly impregnated on the strawberry surface; it is statistically independent of the immersion time. The nanocoating allowed the shelf life of strawberries to be increased at room temperature (25 °C) and at refrigeration temperature (5 °C). The homogeneous distribution of the nanostructured ZnO into the chitosan matrix favored the physicochemical and textural properties of strawberries, including for up to 8 days. The coating with nanostructured ZnO improved the antibacterial properties of chitosan, i.e., a synergistic effect between these two compounds was observed, reducing the microbial load. For strawberries stored at 5 °C, treatment FRBN-1 exhibited the highest moisture content on day 1 (93.2%) compared to the control (91.9%). Additionally, on day 8, treatment FRBN-1 (92.9%) also maintained a high moisture content compared to the control (89.3%). In contrast, for strawberries stored at 25 °C, on day 1, treatment FRCh-1 (82.0%) had the lowest moisture content, significantly different from the control (89.1%) and other treatments. However, on day 8, a significant improvement was observed in treatment FRCh-1 (69.8%), while the control (45.3%) experienced a significant decrease in moisture content compared to day 1. Moreover, results of pH, TSS, and TA are comparable with the parameters of both quality and commercial acceptability of fresh strawberries (pH = 3.17–3.98; TA = 0.60–0.82%; TSS = 6.7–10.3 °Bx) as well as acceptable sensory properties, which suggest the potential of the Ch/ZnO-NPs coating in improving the preservation of fresh strawberries.

## Figures and Tables

**Figure 1 polymers-15-03772-f001:**
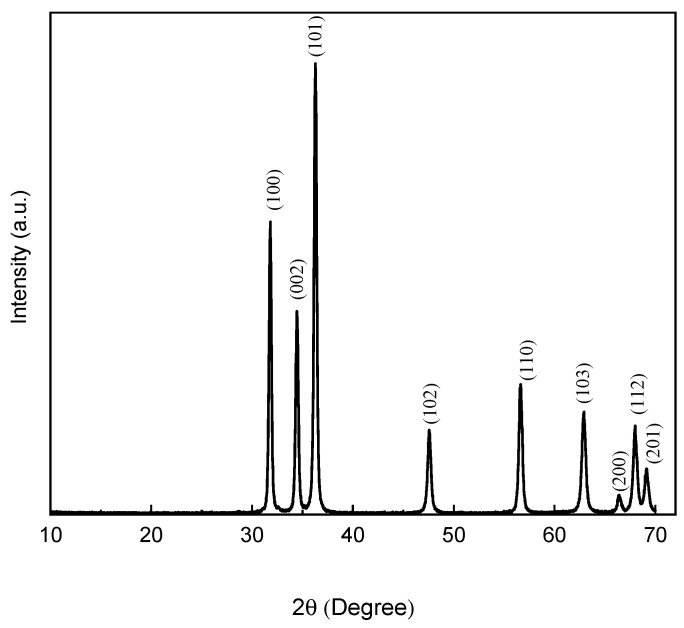
X-ray diffraction pattern of spaghetti-like ZnO nanoparticles.

**Figure 2 polymers-15-03772-f002:**
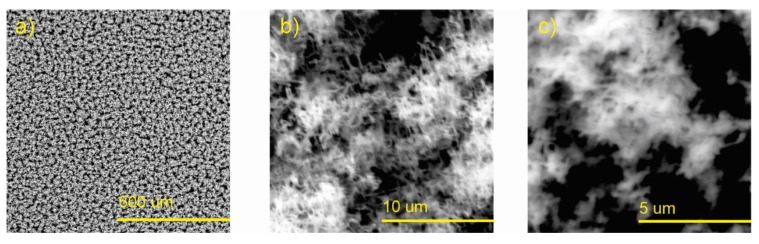
SEM images of the as-prepared Zn nanostructured films by the CSS technique: (**a**) morphology of the zinc spaghetti-like nanostructures at low magnification; (**b**,**c**) correspond to higher magnifications.

**Figure 3 polymers-15-03772-f003:**
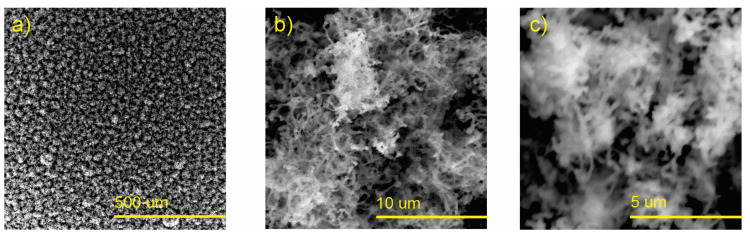
SEM images of the ZnO nanostructured films: (**a**) morphology of the spaghetti-like ZnO nanostructures after thermal annealing at low magnification; (**b**,**c**) are the micrographs at higher magnifications.

**Figure 4 polymers-15-03772-f004:**
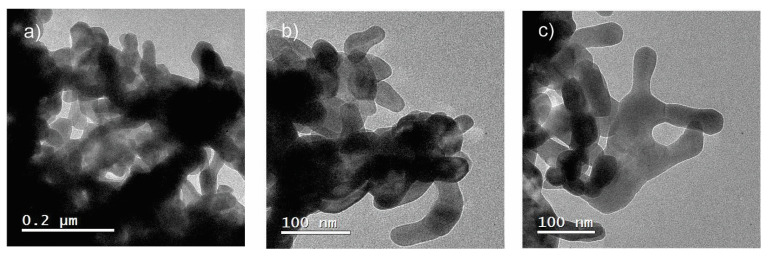
TEM images on the surface of the spaghetti-like ZnO nanostructures at different scale bars: (**a**) 0.2 μm, (**b**,**c**) 100 nm.

**Figure 5 polymers-15-03772-f005:**
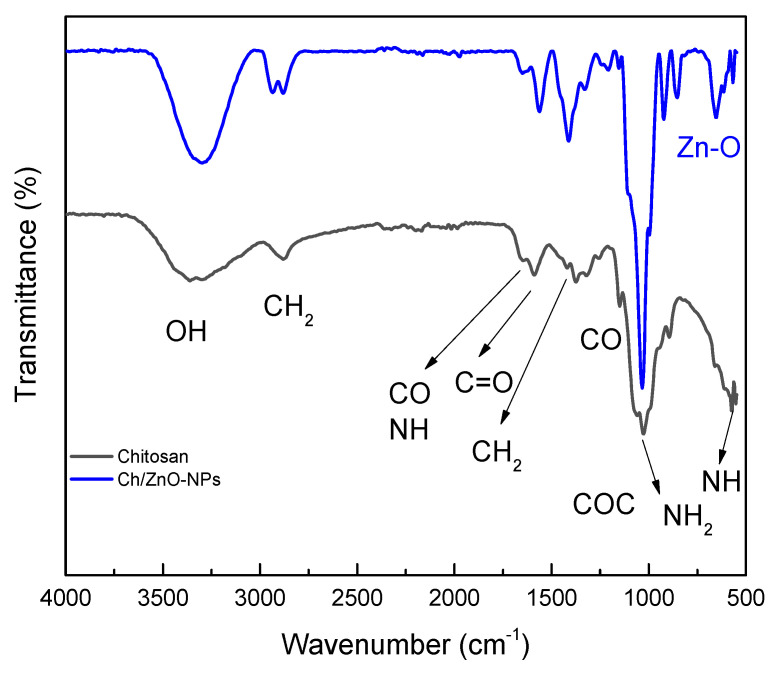
FTIR spectrum of chitosan and of the Ch/ZnO-NPs nanobiocomposite.

**Figure 6 polymers-15-03772-f006:**
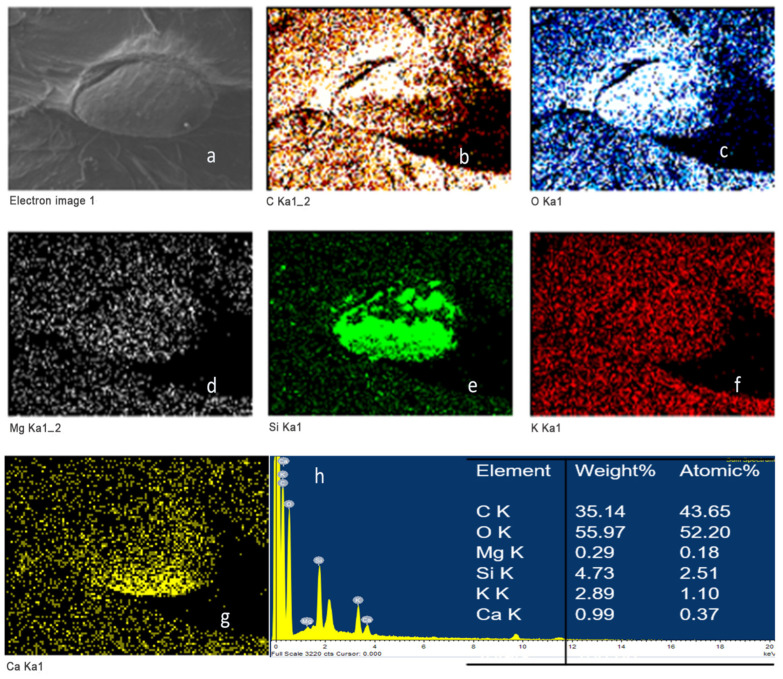
Elemental mapping of strawberries coated with chitosan after one minute of immersion (FRCh-1). (**a**) SEM micrograph, (**b**) carbon, (**c**) oxygen, (**d**) magnesium, (**e**) silicon, (**f**) potassium, (**g**) calcium, and (**h**) Quantitative elemental analysis of the coating.

**Figure 7 polymers-15-03772-f007:**
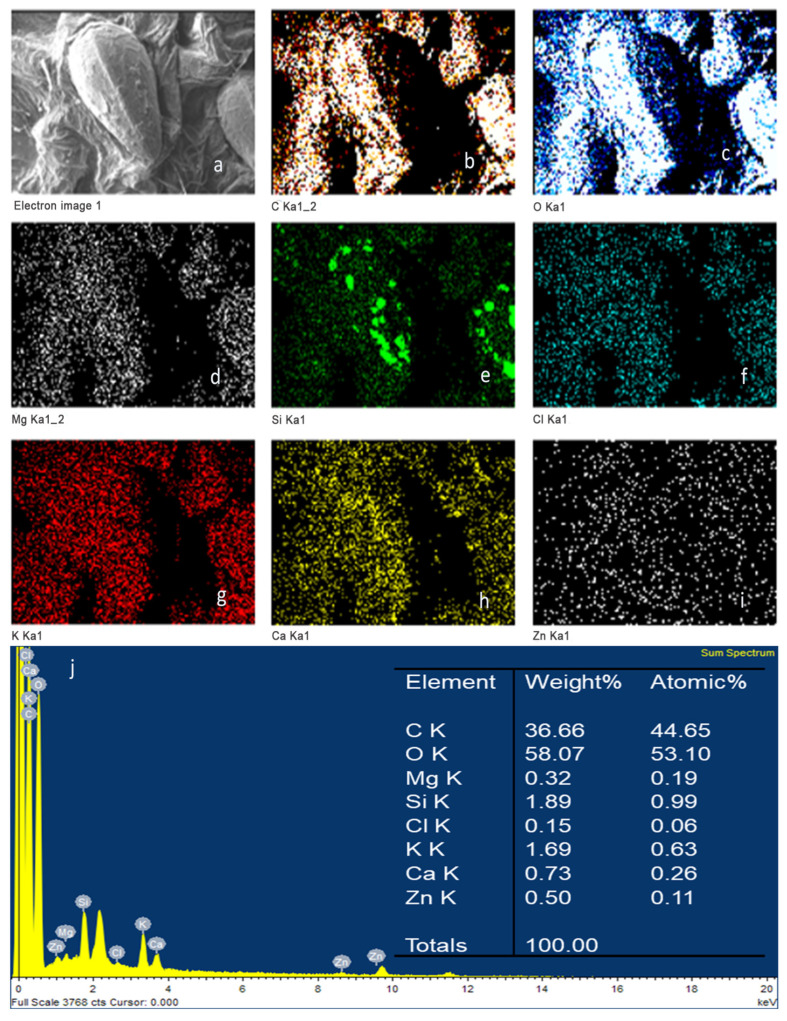
Elemental mapping of the bionanocomposite coated strawberry after one minute of immersion (FRBN-1). (**a**) SEM micrograph obtained, (**b**) carbon, (**c**) oxygen, (**d**) magnesium, (**e**) silicon, (**f**) chlorine, (**g**) potassium, (**h**) calcium, (**i**) zinc, and (**j**) quantitative elemental analysis of the bionanocomposite coating.

**Figure 8 polymers-15-03772-f008:**
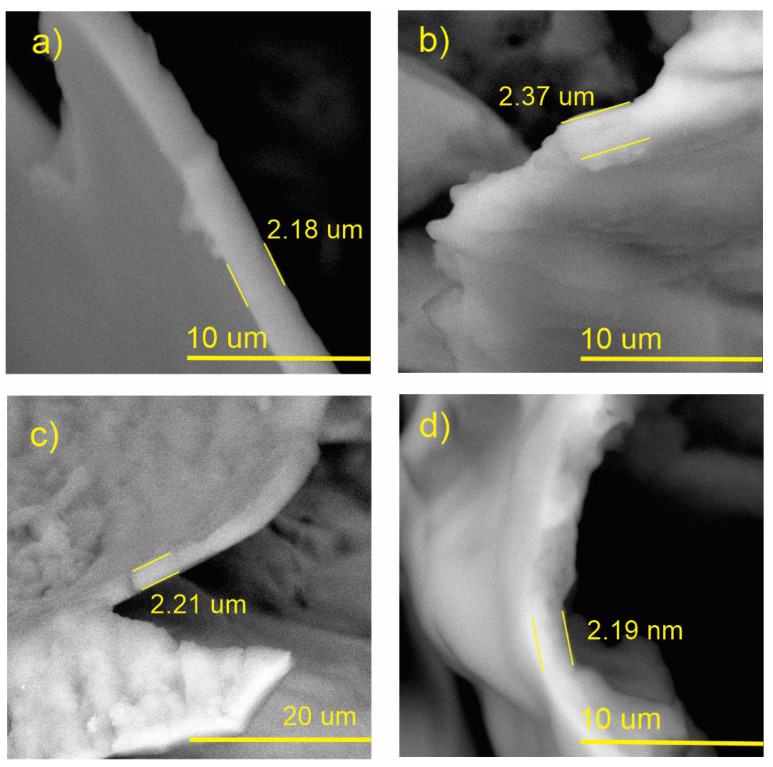
SEM images used for measuring the coating thickness. (**a**) FRCh-1; (**b**) FRCh-2; (**c**) FRBN-1; and (**d**) FRBN-2. 1 h.

**Figure 9 polymers-15-03772-f009:**
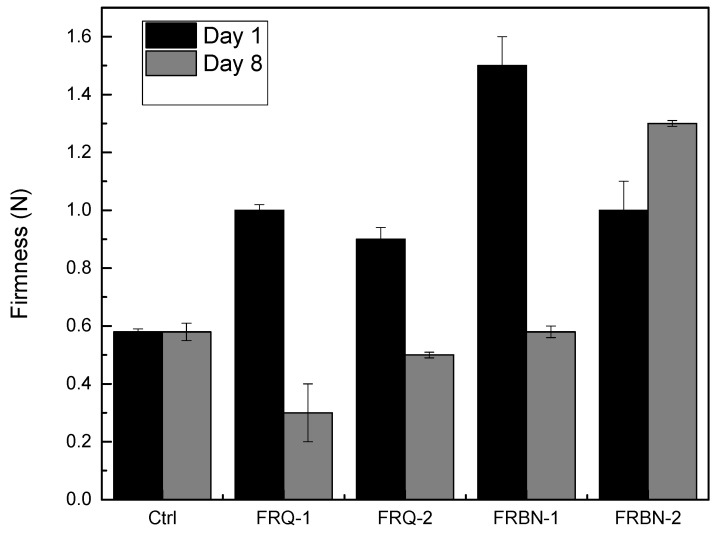
Firmness changes of coated and control strawberries stored at 5 °C for 1 and 8 days. Each data point are the mean of three replicates.

**Table 1 polymers-15-03772-t001:** Coated and uncoated strawberries stored at 5 °C for 8 days.

	Ctrl	FRCh-1	FRBN-1
Day 1	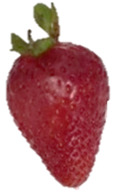	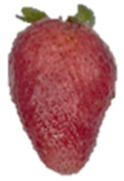	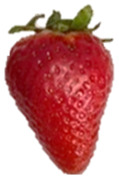
Day 8	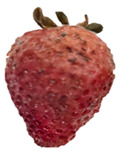	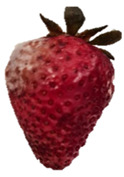	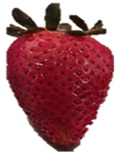

**Table 2 polymers-15-03772-t002:** Coated and uncoated strawberries stored at 25 °C for 8 days.

	Ctrl	FRCh-1	FRBN-1
Day 1	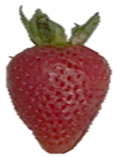	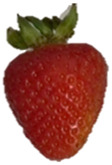	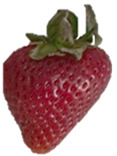
Day 8	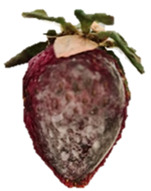	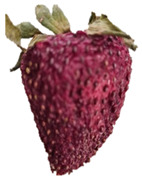	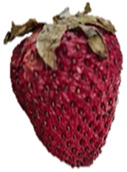

**Table 3 polymers-15-03772-t003:** Effect of the different coatings on the microbiological changes of the treated strawberries stored at 5 and 25 °C.

Strawberry	t_s_ (Days)	T (°C)	AMB (CFU/g)	Molds and Yeasts (CFU/g)
Ctrl	1	5	6.0 × 10^2^ ± 112 ^aC^	1.5 × 10^2^ ± 10 ^aB^
		25	2.0 × 10^2^ ± 12 ^bA^	3.0 × 10^2^ ± 2.6 ^aA^
	8	5	INC	INC
		25	4.0 × 10^3^ ± 22 ^bB^	1.04 × 10^5^ ± 1.6 ^aB^
FRCh-1	1	5	50 ± 1.2 ^aA^	0 ± 0.00 ^aA^
		25	2.0 × 10^2^ ± 7.2 ^bB^	0 ± 0.00 ^aA^
	8	5	1.3 × 10^6^ ± 15,612 ^aB^	3.9 × 10^6^ ± 13,212 ^bB^
		25	4.3 × 10^3^ ± 182 ^bC^	2.9 × 10^4^ ± 1122 ^cC^
FRCh-2	1	5	3.0 × 10^2 aB^	0 ± 0.00 ^aA^
		25	0 ^bC^	0 ± 0.00 ^aA^
	8	5	1.3 × 10^6 aA^	3.9 × 10^6^ ± 10,100 ^bB^
		25	2.4 × 10^3 bA^	3.4 × 10^3^ ± 123 ^cC^
* FRBN-1	1	5	0 ± 0.00 ^aA^	0 ± 0.00 ^aA^
		25	0 ± 0.00 ^aA^	0 ± 0.00 ^aA^
	8	5	7.9 × 10^5^ ± 600 ^bC^	1.2 × 10^6^ ± 1200 ^bB^
		25	0 ± 0.00 ^aA^	0 ± 0.00 ^Aa^
FRBN-2	1	5	0 ± 0.00 ^aA^	0 ± 0.00 ^aA^
		25	0 ± 0.00 ^aA^	0 ± 0.00 ^aA^
	8	5	6.3 × 10^5^ ± 0.00 ^bC^	1.2 × 10^6^ ± 1300 ^bB^
		25	0 ± 0.00 ^aA^	0 ± 0.00 ^aA^

Data are presented as mean ± standard deviation (SD). a, b, c superscripts in the same row indicate a significant difference between the means (*p* ≤ 0.05) of temperature. A, B, C superscripts in the same row indicate a significant difference between the means (*p* ≤ 0.05) of storage time. * Represents significant difference between the means (*p* ≤ 0.05) of the treatments. INC: countless.

**Table 4 polymers-15-03772-t004:** Moisture content of coated and uncoated strawberries stored at different temperatures for 8 days.

t_s_ (Days)	T (°C)	Moisture (%)				
		Ctrl	* FRCh-1	FRCh-2	FRBN-1	FRBN-2
1	5	91.9 ± 2.0 ^aA^	92.6 ± 2.2 ^aB^	92.2 ± 0.9 ^aAB^	93.2 ± 1.1 ^Aa^	91.6 ± 1.3 ^aAB^
	25	89.1 ± 0.6 ^aA^	89.0 ± 1.3 ^aA^	82.0 ± 9.9 ^aB^*	89.0 ± 2.2 ^aA^	92.7 ± 4.0 ^aAC^
8	5	89.3 ± 1.1 ^aA^	92.3 ± 1.4 ^aA^	91.4 ± 1.0 ^aA^	92.9 ± 0.8 ^Aa^	91.0 ± 1.4 ^aA^
	25	45.3 ± 3.7 ^bA^*	69.8 ± 1.5 ^bB^	67.4 ± 2.7 ^bB^	83.0 ± 1.6 ^Bc^	84.5 ± 3.3 ^bC^

Data are presented as mean ± standard deviation (SD). a, b, c superscripts in the same row indicate a significant difference between the means (*p* ≤ 0.05) of the treatments. A, B, C Different superscripts in the same row indicate a significant difference between the means (*p* ≤ 0.05) of storage time. * Represents a significant difference between the means (*p* ≤ 0.05) of the treatments.

**Table 5 polymers-15-03772-t005:** Physicochemical parameters of coated and uncoated strawberries stored at different temperatures for 8 days.

Strawberry	t_s_ (Days)	T (°C)	pH	TSS (°Bx)	TA(%)
Ctrl	1	5	3.59 ± 0.18 ^aA^	8.5 ± 0.2 ^aA^	0.76 ± 0.04 ^aA^
		25	3.17 ± 0.03 ^bB^	8.1 ± 0.1 ^aA^	0.80 ± 0.02 ^aA^
	8	5	3.31 ± 0.01 ^aC^	8.0 ± 0.1 ^aA^	0.76 ± 0.04 ^aA^
		25	3.68 ± 0.03 ^bA^	7.7 ± 0.2 ^bA^	0.60 ± 0.01 ^cB^
* FRCh-1	1	5	3.47 ± 0.02 ^aA^	8.1 ± 0.1 ^aA^	0.77 ± 0.05 ^aA^
		25	3.38 ± 0.08 ^aC^	8.2 ± 0.2 ^aA^	0.79 ± 0.02 ^aA^
	8	5	3.32 ± 0.01 ^aC^	8.0 ± 0.1 ^aA^	0.61 ± 0.01 ^aA^
		25	3.76 ± 0.01 ^bB^	6.9 ± 0.2 ^bB^	0.69 ± 0.03 ^bA^
FRCh-2	1	5	3.53 ± 0.06 ^aA^	8.5 ± 0.5 ^aA^	0.77 ± 0.03 ^aA^
		25	3.31 ± 0.01 ^bC^	7.8 ± 0.3 ^bA^	0.80 ± 0.01 ^aA^
	8	5	3.67 ± 0.06 ^cA^	8.5 ± 0.3 ^aA^	0.60 ± 0.01 ^bB^
		25	3.80 ± 0.18 ^cB^	6.7 ± 1.0 ^bB^	0.68 ± 0.02 ^cA^
* FRBN-1	1	5	3.48 ± 0.01 ^aA^	9.1 ± 0.2 ^cA^	0.80 ± 0.04 ^aCA^
		25	3.37 ± 0.01 ^aC^	9.5 ± 0.3 ^aCA^	0.82 ± 0.02 ^aCA^
	8	5	3.61 ± 0.01 ^bA^	8.6 ± 0.2 ^bA^	0.70 ± 0.01 ^bA^
		25	3.85 ± 0.02 ^aB^	8.5 ± 0.2 ^Ba^	0.77 ± 0.02 ^abA^
FRBN-2	1	5	3.22 ± 0.02 ^aB^	8.9 ± 0.1 ^aA^	0.80 ± 0.01 ^aA^
		25	3.28 ± 0.02 ^aB^	10.3 ± 0.5 ^bCA^	0.80 ± 0.01 ^aA^
	8	5	3.54 ± 0.05 ^bA^	8.3 ± 0.3 ^aA^	0.69 ± 0.01 ^bA^
		25	3.98 ± 0.02 ^aC^	9.2 ± 0.5 ^bC^	0.75 ± 0.04 ^abA^

Data are presented as mean ± standard deviation (SD). a, b, c superscripts in the same row indicate a significant difference between the means (*p* ≤ 0.05) of temperature. A, B, and C superscripts in the same row indicate a significant difference between the means (*p* ≤ 0.05) of storage time. * Represents significant difference between the means (*p* ≤ 0.05) of the treatments.

**Table 6 polymers-15-03772-t006:** Color parameters of coated and uncoated strawberries stored at 5 and 25 °C after 8 days of storage.

Treatment	t_s_ (Days)	T (°C)	*L*	*a**	*b**	Browning Index
Ctrl	1	5	18.21 ± 0.87 ^aA^	20.48 ± 2.43 ^aA^	6.01 ± 1.08 ^aA^	108.9 ± 11.1 ^aA^
		25	22.42 ± 3.42 ^bC^	22.28 ± 0.95 ^aC^	6.57 ± 0.47 ^bA^	97.7 ± 8.70 ^bB^
	8	5	22.18 ± 3.70 ^bC^	17.22 ± 1.10 ^aD^	7.08 ± 1.17 ^aA^	89.6 ± 8.30 ^aB^
		25	20.05 ± 8.35 ^bB^	08.01 ± 3.74 ^bE^	3.95 ± 0.41 ^bB^	56.0 ± 27.2 ^bC^
FRCh-1	1	5	18.57 ± 4.00 ^aA^	20.26 ± 5.10 ^aA^	5.87 ± 2.69 ^aC^	104.0 ± 13.5 ^aA^
		25	18.83 ± 1.43 ^aA^	17.71 ± 2.58 ^bD^	6.40 ± 2.83 ^bA^	101.2 ± 22.2 ^aA^
	8	5	22.17 ± 1.63 ^bB^	21.78 ± 3.42 ^aB^	11.88 ± 10.29 ^aD^	148.5 ± 99.1 ^aD^
		25	18.42 ± 1.97 ^aA^	13.36 ± 2.14 ^bE^	4.34 ± 0.91 ^bC^	73.6 ± 5.8 ^bEC^
FRCh-2	1	5	19.59 ± 0.61 ^aAB^	22.41 ± 0.88 ^aC^	6.96 ± 1.12 ^aA^	114.0 ± 7.5 ^aA^
		25	21.65 ± 1.31 ^bBC^	19.55 ± 2.20 ^aD^	6.25 ± 1.29 ^aA^	91.1 ± 8.8 ^bB^
	8	5	20.32 ± 3.13 ^aB^	49.03 ± 1.8 ^aF^	11.97 ± 10.35 ^aD^	152.1 ± 88.2 ^aD^
		25	15.66 ± 1.29 ^bD^	37.78 ± 0.86 ^bG^	2.50 ± 0.67 ^bB^	50.6 ± 3.2 ^Bc^
FRBN-1	1	5	20.36 ± 3.10 ^aAB^	22.06 ± 1.40 ^aC^	6.84 ± 1.18 ^aA^	108.4 ± 6.2 ^aA^
		25	19.30 ± 2.38 ^aAB^	17.81 ± 3.90 ^bD^	5.17 ± 1.54 ^aAC^	88.6 ± 12.5 ^bB^
	8	5	21.71 ± 5.35 ^aBC^	61.88 ± 1.9 ^aH^	5.68 ± 2.41 ^aAC^	93.9 ± 5.4 ^aB^
		25	17.50 ± 2.50 ^bD^	11.04 ± 3.5 ^bI^	3.31 ± 0.88 ^Bb^	61.5 ± 9.2 ^bC^
FRBN-2	1	5	20.12 ± 1.32 ^aB^	24.32 ± 1.17 ^aC^	7.15 ± 0.40 ^Aa^	117.4 ± 4.6 ^aA^
		25	20.89 ± 1.65 ^bB^	18.21 ± 0.77 ^bD^	5.74 ± 1.15 ^bAC^	87.6 ± 4.0 ^bB^
	8	5	21.88 ± 2.07 ^aBC^	21.88 ± 1.91 ^aC^	7.31 ± 1.44 ^aA^	103.1 ± 5.1 ^aA^
		25	19.00 ± 2.03 ^Bab^	12.07 ± 1.27 ^bE^	8.82 ± 0.34 ^bA^	124.6 ± 103.2 ^aA^

Data are presented as mean ± standard deviation (SD). a, b, c superscripts in the same row indicate a significant difference between the means (*p* ≤ 0.05) of temperature. A, B, C, D, E, F, G, H superscripts in the same row indicate a significant difference between the means (*p* ≤ 0.05) of storage time.

## Data Availability

The data presented in this study are available on request from the corresponding author.

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
