# Peer review of "Chitosan Coatings Modified with Nanostructured ZnO for the Preservation of Strawberries"

_polymers, 2023, doi:10.3390/polym15183772_

Round 1
Reviewer 1 Report
Overview and general recommendation:
Approximately 1.3 billion tons of food are wasted worldwide each year, with perishable fruits and vegetables contributing to around 40% of this total. This highlights the significance of developing sustainable solutions to prolong food shelf life and combat food waste, as it can have a substantial global impact.
Regarding the study presented by the authors, I felt confident that they conducted a meticulous and comprehensive investigation. However, I also observed that certain essential aspects were inadequately described or entirely omitted. In the following sections, I will elaborate on my specific concerns in more detail.
Comments:
In the Materials and Methods section, it really should be mentioned the accelerating voltages that were used in morphological characterizations and EDX measurements of the samples. The EDX data were obtained from how many different locations on each surface (…x magnification/location), and mediated? Provide sufficient details to allow the work to be reproduced by an independent researcher.
The usage of references within round brackets in the Results section, particularly in the XRD sub-section, can be quite confusing.
Figure 2 and 3 would benefit from improvement, such as removing the "down" label and including only the scale on the figure to enhance their clarity and identification for the reader. The captions should focus on the scale rather than magnifications, as it is more essential and beneficial for the reader. Additionally, the statement "their lengths reach approximately 5 μm, with cross-section between 100 and 300 nm (Figure 2b, 2c)" should be proved by indicating directly on the figure.
The analysis of the EDX spectrum is somewhat unreliable due to line overlap, especially for elements like C, N, O, etc., where the spectral lines fall beyond the instrumental resolution range. While the standardless EDS technique used is semi-quantitative, considering your objective to observe elemental uniformity, it is advisable to obtain EDX spectra from a minimum of three regions (with magnifications of 500x or 1000x for each location) and then compare the results.
The statement "Results showed that, in addition to C (35.70%) and O (54.77%), Zn (0.56 %) was found uniformly distributed on the strawberry surface" might be based on elemental mapping. In that case, the elemental mapping needs improvement, as it is not evident from Figure 7 i) and should be the most crucial figure for illustrating this finding.
The statement "The ZnO nanostructures on the coating have a uniform granular morphology with small agglomerations, which indicates a homogenous dispersion and integration into the chitosan matrix" lacks specific information about the result that led to this conclusion. Further clarification or data presentation is needed to support this statement.
It’s recommended to enhance the quality and reliability of the EDX spectra in Figure 6 and 7. The atomic/weight % are presented with the elimination of the background? Because from the spectra it can be seen that the background subtraction of EDX spectrum was not used in software.
In the "3.6. Coating thickness" section, it would be beneficial to include references to specific figures, such as Figure 8 a) or b), when making relevant statements. For instance, when mentioning "the appearance of randomly distributed particles on the surface of the strawberry was observed," it is essential to explicitly indicate the location of these particles on the figure, using arrows or other markers to guide the reader's attention to the relevant areas. This will ensure that readers can easily visualize and understand the observations being described. To enhance the figures, consider removing unnecessary labels and including only the scale for better clarity and presentation. By eliminating the labels and incorporating a clear scale indicator, the figures will become more reader-friendly and easier to interpret.
Your research presents a valuable contribution to the field, and the findings are intriguing. However, in order to enhance the impact of your study, I kindly request you to provide the biocompatibility of materials as it is essential for their biomaterial application. It is essential to determine the cytotoxicity of the prepared coatings using the MTT assay, and the cell survival percentage on the samples to demonstrate its suitable biocompatibility. This could be attributed to the fact that the prepared sample is a harmless and biocompatible composite.
Minor editing of English language required.
Author Response
Professor William Faccinatto
Editor
Special issue Research Progress on Chitosan Applications
Regarding the paper “Chitosan modified with nanostructured ZnO coating for the preservation of strawberries” (Polymers-2550980) Dulce J. García-García, G. F. Pérez-Sánchez, H. Hernández-Cocoletzi, M. G. Sánchez-Arzubide, M.L. Luna-Guevara, E. Rubio-Rosas, K. Rambabu, C. Morán-Raya, submitted to the Journal Polymers (Special issue Research Progress on Chitosan Applications), I inform you that we have addressed the reviewers comments. The revised manuscript has been uploaded and the changes highlighted. We appreciate the reviewers comments and they have been addressed as follow:
The title has been modified to “Chitosan coatings modified with nanostructured ZnO for the preservation of strawberries”.
Reviewer 1
Approximately 1.3 billion tons of food are wasted worldwide each year, with perishable fruits and vegetables contributing to around 40% of this total. This highlights the significance of developing sustainable solutions to prolong food shelf life and combat food waste, as it can have a substantial global impact.
Regarding the study presented by the authors, I felt confident that they conducted a meticulous and comprehensive investigation. However, I also observed that certain essential aspects were inadequately described or entirely omitted. In the following sections, I will elaborate on my specific concerns in more detail.
Comments:
- In the Materials and Methods section, it really should be mentioned the accelerating voltages that were used in morphological characterizations and EDX measurements of the samples. The EDX data were obtained from how many different locations on each surface (…x magnification/location), and mediated? Provide sufficient details to allow the work to be reproduced by an independent researcher.
Response:
We appreciate the comments. The accelerating voltage used was of 20 kV, in both, morphological characterizations and EDX measurements; all the image field was considered. The mapping was developed in a 3x3 area and in 3 different zones. This was indicated and highlighted in section 2.5.
- The usage of references within round brackets in the Results section, particularly in the XRD sub-section, can be quite confusing.
Response:
We appreciate the comment. The round brackets have been changed accordingly
- Figure 2 and 3 would benefit from improvement, such as removing the "down" label and including only the scale on the figure to enhance their clarity and identification for the reader. The captions should focus on the scale rather than magnifications, as it is more essential and beneficial for the reader. Additionally, the statement "their lengths reach approximately 5 μm, with cross-section between 100 and 300 nm (Figure 2b, 2c)" should be proved by indicating directly on the figure.
Response:
We agree with the comment, and appreciate it. Figures 2 and 3 have been modified accordingly. The corresponding figure caption has been written appropriately.
- The analysis of the EDX spectrum is somewhat unreliable due to line overlap, especially for elements like C, N, O, etc., where the spectral lines fall beyond the instrumental resolution range. While the standardless EDS technique used is semi-quantitative, considering your objective to observe elemental uniformity, it is advisable to obtain EDX spectra from a minimum of three regions (with magnifications of 500x or 1000x for each location) and then compare the results.
Response:
Dear reviewer, we sincerely appreciate your feedback, which likely stems from an error we unfortunately made. When transcribing the elemental chemical analysis in the figure 6 (h), we overlooked a crucial detail: nitrogen should be replaced by oxygen. This adjustment ensures that the content aligns accurately with the analysis presented in figure 7 (j). We want to inform you that we have promptly corrected this mistake.
- The statement "Results showed that, in addition to C (35.70%) and O (54.77%), Zn (0.56 %) was found uniformly distributed on the strawberry surface" might be based on elemental mapping. In that case, the elemental mapping needs improvement, as it is not evident from Figure 7 i) and should be the most crucial figure for illustrating this finding.
Response:
Thank you for your observation. The text has been changed as follow “Additional to C (35.70%) and O (54.77%), a homogeneous distribution of Zn (0.56 %) (Figure 7i) was found on the strawberry surface without severe aggregation 1e (25,35)”. We have adjusted the color and intensity of the chemical mapping for the zinc element in the figure 7i. We are confident that this change will allow for a clearer appreciation of the zinc distribution.
- The statement "The ZnO nanostructures on the coating have a uniform granular morphology with small agglomerations, which indicates a homogenous dispersion and integration into the chitosan matrix" lacks specific information about the result that led to this conclusion. Further clarification or data presentation is needed to support this statement.
Response:
We agree with the reviewer; it is observed from the mapping that the Zn element is uniformly distributed on the strawberry surface, then, we may infer that also, the ZnO nanostrustrues are uniformly distributed on the Surface. The text has been changed to “The homogeneous distribution of the Zn, as shown in figure 7i, permits to infer that the ZnO nanostructures are uniformly distributed on the coating, which indicates a homogenous dispersion and integration into the chitosan matrix”
- It’s recommended to enhance the quality and reliability of the EDX spectra in Figure 6 and 7. The atomic/weight % are presented with the elimination of the background? Because from the spectra it can be seen that the background subtraction of EDX spectrum was not used in software.
Response:
We thanks the comment. To clarify more, the EDX spectra has been modified and eliminated the Au signal.
- In the "3.6. Coating thickness" section, it would be beneficial to include references to specific figures, such as Figure 8 a) or b), when making relevant statements. For instance, when mentioning "the appearance of randomly distributed particles on the surface of the strawberry was observed," it is essential to explicitly indicate the location of these particles on the figure, using arrows or other markers to guide the reader's attention to the relevant areas. This will ensure that readers can easily visualize and understand the observations being described. To enhance the figures, consider removing unnecessary labels and including only the scale for better clarity and presentation. By eliminating the labels and incorporating a clear scale indicator, the figures will become more reader-friendly and easier to interpret.
Response:
We appreciate the reviewer comment.
Figure 8 has been improved including the scale bars as well as an appropriate image
- Your research presents a valuable contribution to the field, and the findings are intriguing. However, in order to enhance the impact of your study, I kindly request you to provide the biocompatibility of materials as it is essential for their biomaterial application. It is essential to determine the cytotoxicity of the prepared coatings using the MTT assay, and the cell survival percentage on the samples to demonstrate its suitable biocompatibility. This could be attributed to the fact that the prepared sample is a harmless and biocompatible composite.
Response:
We appreciate the reviewer comment.
It has been demonstrated that chitosan is a biomaterial, which has been empoyed even in the tissue engineering. At the same time, ZnO nanoparticles are also used for their antibacterial activity; then, it is expected that the coated obtained in this work is biocompatible. We agree that citotoxicity studies are important for these kind of applications; however, we are no table to conduct this studies and include them in the present manuscript, they will be developed in a next work.
Reviewer 2 Report
1. Line 14: what properties of strawberries are so crucial for human life? [revise the sentence, it is overestimated]
2. Lines 32-39 - Please refer rather to the strawberries storage instead of general statements
3. Line 44: "nano-ZnO"
4. Line 47: is there any limit on ZnO consumption? The introduction of ZnO on the GRAS list was made according to the report made in 1973 (50 years ago). It was marked as "2" (i.e., There is no evidence in the available information on [substance] that demonstrates a hazard to the public when it is used at levels that are now current and in the manner now practiced. However, it is not possible to determine, without additional data, whether a significant increase in consumption would constitute a dietary hazard. ), thus, it refers only to the materials used 50 years ago. Please refer to the current possible consumption limit or any research devoted to the possible consumption of ZnO currently.
5. Please describe what was done after removing strawberries from the coating solution. As the excess of the coating can freely run down the fruit surface, the final coating thickness can be affected by this process.
6. Eq. 1 - remove "("
7. Describe TEM analysis in the methodology section
8. Add Figure 4 caption. Moreover, a, b, and c notation is not visible
9. Lines 251-252 - why do Authors refer to the starch structure?
10. Line 267: the table in Figure 6 indicates the main elements (C - 33.78% and N (not O) - 52,51%, there is a lack of info regarding the suggested O percentage)
11. The sharpness of the images in Fig. 8 should be improved. The SEM image quality is not enough to be able to measure the polymer coating thickness. Moreover, it is hard to distinguish the fruit and the coating boundary.
12. Lines 313-315: sentence needs revision
13. What are the visible differences in strawberries (Table 1 and 2) between FRCh-1 and FRBN-1 - is there really a difference in the fungi presence between neat Ch and ZnO/Ch coatings? Please provide images with higher fruit images and better resolution.
14. Table 3 and 5 is hard to read - please divide it into sections. I could not even refer to this data
15. Line 182 stated that p> 0.05 for all samples, while data in Tables 3 and 4 are indicated as p <0.05. Then in line 419 again, p<0.05 indicate no statistically different result, and inversely line 449. Please correct the discussion of variance.
16. Why are firmness data shown only at 5 (not 25?)
17. There is no Table 6 in the text
18. Indicate in conclusions what parameters were improved - give specific values or differences in %
-
Author Response
Professor William Faccinatto
Editor
Special issue Research Progress on Chitosan Applications
Regarding the paper “Chitosan modified with nanostructured ZnO coating for the preservation of strawberries” (Polymers-2550980) Dulce J. García-García, G. F. Pérez-Sánchez, H. Hernández-Cocoletzi, M. G. Sánchez-Arzubide, M.L. Luna-Guevara, E. Rubio-Rosas, K. Rambabu, C. Morán-Raya, submitted to the Journal Polymers (Special issue Research Progress on Chitosan Applications), I inform you that we have addressed the reviewers comments. The revised manuscript has been uploaded and the changes highlighted. We appreciate the reviewers comments and they have been addressed as follow:
The title has been modified to “Chitosan coatings modified with nanostructured ZnO for the preservation of strawberries”.
Reviewer 2
- Line 14: what properties of strawberries are so crucial for human life? [revise the sentence, it is overestimated]
Response: We appreciate the comment. The sentence has been changed to “Strawberries are highly consumed around the world”
Lines 32-39 - Please refer rather to the strawberries storage instead of general statements
Response: We appreciate the comment. All the first paragraph has been modified accordingly.
Line 44: "nano-ZnO"
Response: "nano-ZnO" has been written appropriately.
Line 47: is there any limit on ZnO consumption? The introduction of ZnO on the GRAS list was made according to the report made in 1973 (50 years ago). It was marked as "2" (i.e., There is no evidence in the available information on [substance] that demonstrates a hazard to the public when it is used at levels that are now current and in the manner now practiced. However, it is not possible to determine, without additional data, whether a significant increase in consumption would constitute a dietary hazard. ), thus, it refers only to the materials used 50 years ago. Please refer to the current possible consumption limit or any research devoted to the possible consumption of ZnO currently.
Response: we appreciate the comment. Additional studies are required to confirm the nano ZnO safety. We have actualized reference 5 and included an additional paragraph witj a reference.
Please describe what was done after removing strawberries from the coating solution. As the excess of the coating can freely run down the fruit surface, the final coating thickness can be affected by this process.
Response: we appreciate the comment. The coating was adhered to strawberry surface and the runoff was insignificant. A comment was included in section 2.4.
Eq. 1 - remove "("
Response: we appreciate the observation, "(" has been removed.
Describe TEM analysis in the methodology section
Response: we appreciate the comment. The procedure for the TEM analysis was described in section 2.3.
Add Figure 4 caption. Moreover, a, b, and c notation is not visible
Response: we appreciate the comment. Figure 4 has been improved.
Lines 251-252 - why do Authors refer to the starch structure? checar
Line 267: the table in Figure 6 indicates the main elements (C - 33.78% and N (not O) - 52,51%, there is a lack of info regarding the suggested O percentage)
Response: We appreciate the reviewer comment. Figure 6 has been changed accordingly.
- The sharpness of the images in Fig. 8 should be improved. The SEM image quality is not enough to be able to measure the polymer coating thickness. Moreover, it is hard to distinguish the fruit and the coating boundary.
Response: The SEM images have been corrected and re-edited; in section 3.6 it has been specified that only coating thicknesses are discussed.
Lines 313-315: sentence needs revisión
Response: authors appreciate the comment. The sentence has been revised.
What are the visible differences in strawberries (Table 1 and 2) between FRCh-1 and FRBN-1 - is there really a difference in the fungi presence between neat Ch and ZnO/Ch coatings? Please provide images with higher fruit images and better resolution.
Response: we appreciate the comment. Visually, the presence of mold, the generation of fermentative odors, and the loss of turgidity and firmness were observed in strawberries subjected to both temperatures on both day 1 and day 8 for the FRCh-1 treatment, but not for the FRBN-1 treatment.
Yes, there is indeed a difference in the presence of fungi between the neat Ch (chitosan) and ZnO/Ch (zinc oxide/chitosan) coatings when comparing FRCh-1 and FRBN-1 treatments. The presence of fungi is more pronounced in the neat Ch coating (FRCh-1) compared to the ZnO/Ch coating (FRBN-1). This suggests that the incorporation of zinc oxide (ZnO) nto the chitosan coating (ZnO/Ch) has a positive effect in inhibiting the presence of fungi on the strawberries. This observation indicates that the ZnO/Ch coating is more effective in preventing fungal growth compared to the neat chitosan coating.
This has been incorporated in section 3.6
Table 3 and 5 is hard to read - please divide it into sections. I could not even refer to this data
Response: we appreciate the comment. Lines have been incorporated in tables 3 and 5 ir order to facilitate the read the data.
Line 182 stated that p> 0.05 for all samples, while data in Tables 3 and 4 are indicated as p <0.05. Then in line 419 again, p<0.05 indicate no statistically different result, and inversely line 449. Please correct the discussion of variance.
Response: we appreciate the comment. All the inconsistencies have been addressed.
Why are firmness data shown only at 5 (not 25?)
Response: we appreciate the comment. The samples subjected to storage at room temperature (25 °C) presented a loss of firmness, impeding the measurement; this was indicated in section 3.8
There is no Table 6 in the text
Response: we appreciate the comment. Effectively, Table 6 was missing; it was included in the manuscript.
Indicate in conclusions what parameters were improved - give specific values or differences in %.
Response: we appreciate the coment. A paragraph in the conclusions section have been included.
Reviewer 3 Report
Upon reviewing your manuscript intitled: Chitosan modified with nanostructured ZnO coating for the preservation of strawberries. I find your work interesting, but I do not believe it can be published in its current form. Therefore, some revisions must be done so that it may be published in this journal. I have the following comments and recommendations.
-The introduction of relevant background and research progress was not comprehensive enough.
-It is not clear why this work is important
- In figure 1 the Miller indices must be in parentheses.
- The crystallite sizes for the various samples were calculated by the W-H formulation. However, it is well known that the peak broadening is impacted by other factors such as instrument-related broadening, residual stresses in crystals, etc. How did the authors determine or subtract these well-known parameters?
- References must be in square brackets
- In a general, it is appreciable that the authors have studied. Chitosan modified with nanostructured ZnO coating for the preservation of strawberries, but there is no in-depth scientific discussion. A careful review of the entire manuscript must be performed. To strengthen the discussions, more up-to-date bibliographic references must be sought and included in the text.
Author Response
Professor William Faccinatto
Editor
Special issue Research Progress on Chitosan Applications
Regarding the paper “Chitosan modified with nanostructured ZnO coating for the preservation of strawberries” (Polymers-2550980) Dulce J. García-García, G. F. Pérez-Sánchez, H. Hernández-Cocoletzi, M. G. Sánchez-Arzubide, M.L. Luna-Guevara, E. Rubio-Rosas, K. Rambabu, C. Morán-Raya, submitted to the Journal Polymers (Special issue Research Progress on Chitosan Applications), I inform you that we have addressed the reviewers comments. The revised manuscript has been uploaded and the changes highlighted. We appreciate the reviewers comments and they have been addressed as follow:
The title has been modified to “Chitosan coatings modified with nanostructured ZnO for the preservation of strawberries”.
Reviewer 3
Upon reviewing your manuscript intitled: Chitosan modified with nanostructured ZnO coating for the preservation of strawberries. I find your work interesting, but I do not believe it can be published in its current form. Therefore, some revisions must be done so that it may be published in this journal. I have the following comments and recommendations.
- The introduction of relevant background and research progress was not comprehensive enough.
Response:
We appreciate the reviewer comment.
Introduction has been improved and some additional paragraphs and references were added.
- It is not clear why this work is important.
Response:
We appreciate the reviewer comment.
We have included some paragraphs in the introduction in order to cover this comment.
- In figure 1 the Miller indices must be in parentheses.
Response:
We appreciate the reviewer comment.
The Miller indices have been written in parentheses
- The crystallite sizes for the various samples were calculated by the W-H formulation. However, it is well known that the peak broadening is impacted by other factors such as instrument-related broadening, residual stresses in crystals, etc. How did the authors determine or subtract these well-known parameters?
Response:
We appreciate the reviewer comment
In order to being clear, the paragraph was changed as folow:
The crystalline size of ZnO nanostructures was calculated using Williamson-Hall (W-H) method using the software provided by the XRD system (High Score Plus for Crystallite Size Analysis). The W-H analysis is an integral breadth method where both size and strain-induced broadening are considered in the deconvolution of the peak versus 2θ. It is important to mention that in this research only the crystallite size was considered because W-H plot is more realistic than using the Debye-Scherrer equation, moreover, the result estimated from this analysis was in good agreement with the corresponding HR-TEM results
- References must be in square brackets
Response:
We appreciate the reviewer comment.
References were written in brackets.
- In a general, it is appreciable that the authors have studied. Chitosan modified with nanostructured ZnO coating for the preservation of strawberries, but there is no in-depth scientific discussion. A careful review of the entire manuscript must be performed. To strengthen the discussions, more up-to-date bibliographic references must be sought and included in the text.
Response:
We appreciate the reviewer comment. The entire manuscript has been revised and improved accordingly.
Round 2
Reviewer 1 Report
I am thrilled to communicate that the authors' meticulous attention to addressing the reviewers' feedback and incorporating the requisite revisions has significantly elevated the quality and relevance of your paper titled "Chitosan coatings modified with nanostructured ZnO for the preservation of strawberries."
Reviewer 3 Report
This version can be accepted for publication